# Oligodendrocyte-derived IL-33 functions as a microglial survival factor during neuroinvasive flavivirus infection

**Geoffrey T. Norris[1], Joshua M. Ames[1], Steven F. Ziegler[1,2], Andrew Oberst[1‡]***

**1** Department of Immunology, University of Washington, Seattle Washington, United States of America,
**2** Center for Fundamental Immunology, Benaroya Research Institute, Seattle Washington, United States of America

‡ Lead Contact
* Oberst@UW.edu

**Data Availability Statement:** RNA sequencing data is available via the Gene Expression Omnibus, accession number GSE241804, or via this link:

## Abstract

In order to recover from infection, organisms must balance robust immune responses to pathogens with the tolerance of immune-mediated pathology. This balance is particularly critical within the central nervous system, whose complex architecture, essential function, and limited capacity for self-renewal render it susceptible to both pathogen- and immune-mediated pathology. Here, we identify the alarmin IL-33 and its receptor ST2 as critical for host survival to neuroinvasive flavivirus infection. We identify oligodendrocytes as the critical source of IL-33, and microglia as the key cellular responders. Notably, we find that the IL-33/ST2 axis does not impact viral control or adaptive immune responses; rather, it is required to promote the activation and survival of microglia. In the absence of intact IL-33/ST2 signaling in the brain, neuroinvasive flavivirus infection triggered aberrant recruitment of monocyte-derived peripheral immune cells, increased neuronal stress, and neuronal cell death, effects that compromised organismal survival. These findings identify IL-33 as a critical mediator of CNS tolerance to pathogen-initiated immunity and inflammation.

## Author summary

West Nile virus (WNV) is a mosquito-transmitted virus that can infect the central nervous system (CNS), causing severe encephalitis, long-term cognitive and motor disfunction, and death. This study identifies a mechanism by which WNV infection promotes microglial and host survival independent of viral restriction. Using mouse models, we identify the alarmin IL-33 as a molecule released from dying oligodendrocytes to activate microglia following WNV infection, and we find that mice lacking IL-33 or its receptor ST2 succumb to WNV CNS infection more readily than wild-type controls. Surprisingly, mice lacking IL-33 or ST2 did not show defects in adaptive immune responses or viral clearance. Rather, their increased susceptibility to WNV was characterized by a failure of brain macrophage survival and activation, increased neuronal stress, and increased cell death within the CNS. These findings suggest that IL-33/ST2 signaling promotes disease tolerance to safeguard CNS function during neuroinvasive WNV infection.

https://www.ncbi.nlm.nih.gov/geo/query/acc.cgi?
acc=GSE241804.

**Funding:** This work was supported by NIH/NIAID
grants R01 AI132595 and R21 AI178512 to AO. AO
and GN received salary support from NIH/NIAID.
JMA was supported by the Helen Hay Whitney
Foundation, and received salary support from this
funder. The funders had no role in study design,
data collection and analysis, decision to publish, or
preparation of the manuscript.

**Competing interests:** The authors have declared
that no competing interests exist.

## Introduction

Immune responses within the central nervous system (CNS) must both clear pathogens and
avoid immune-mediated damage to the complex neuronal architecture that makes up this
organ system. As the resident cells immune cells of the CNS, microglia are ideally positioned
to mediate both pathogen elimination and tissue preservation. Microglia are long-lived tissue
resident macrophages unique in their origin entirely from embryonic yolk sac progenitors
with no contribution from other hematopoietic compartments, barring significant CNS disease [1]. Recent advances have described multiple roles for microglia in both CNS homeostasis
and upon its perturbation [2,3]. Recent studies have also revealed mechanisms of microglial
function during CNS viral infection [4], where microglia have been shown to contain infection
and tissue damage during acute infection, while mediating neuronal and tissue damage if aberrantly activated during recovery. These findings highlight the pleotropic nature of microglial
functions in the CNS, and point to a need to better understand the signals that promote the
distinct roles of this cell type.

The cytokine IL-33 is a member of the IL-1 family of alarmin cytokines. IL-33 is normally
released by damaged or necrotic cells, signaling in a paracrine manner to bystanders expressing the IL-33 receptor ST2 (encoded by *Il1rl1)*. IL-33/ST2 signaling has divergent roles in
peripheral immune responses, notably contributing to both immune cell activation and tissue
homeostasis, including roles in promoting macrophage survival [5]. Notably, IL-33 is highly
expressed the CNS, and several roles for IL-33 in CNS development and pathology have been
identified, including promotion of synaptic pruning and extracellular matrix remodeling by
microglia [6,7], and promotion of recovery from spinal cord injury through macrophage signaling [8]. IL-33 signaling has also been identified as a driver of both protective and detrimental CNS immune responses to *Toxoplasma gondii* (where astrocytes are the primary responder
to IL-33 signaling), malaria, Dengue, and Rocio viruses [9–13]. Therefore, the contribution of
IL-33 to beneficial or pathogenic responses to viral infection deserves further characterization.

Since its arrival in the Western Hemisphere, the flavivirus West Nile virus (WNV) has
become a significant threat to public health, with the majority of disease burden resulting from
neuroinvasive infection [14–16]. Recent studies using an attenuated strain of West Nile virus,
termed WNV-E218A, as well as a less neurovirulent form of the related Zika virus (ZIKV-Dakar), have shown bivalent roles for resident microglia in promoting neurological pathology
and cognitive sequelae to flavivirus infection, while also contributing to host survival during
acute infection. [17–20]. Specifically, during chronic infection microglia were shown to be detrimental, phagocytosing neuronal processes in the murine hippocampus, a phenomenon
underlying disease sequelae specific to episodic-spatial memory [21]. However, pharmacological removal of microglia yielded decreased host and neuronal survival, with minimal changes
in viral titer [20], suggesting that microglia play beneficial roles in host survival during neuroinvasive viral infection.

In this study, we show that IL-33 signaling promotes survival during flavivirus infection, an
effect apparently independent of a role for IL-33 in viral control. We found that following CNS
infection IL-33+ oligodendrocytes are lost, and that IL-33 expression by oligodendrocytes is
necessary for host survival, implicating a release of IL-33 from dying oligodendrocytes as the
source of this alarmin. The target of IL-33 signaling was primarily microglia, as mice with specific deletion of ST2 on microglia succumbed more readily to CNS disease. In the absence of
IL-33 signaling post-infection, we found defects in microglial activation and survival, leading
to enhanced BBB breakdown, increased monocyte extravasation and neuronal apoptosis.
Together, our studies reveal a source for CNS IL-33 during flavivirus infection and highlight
its role in promoting microglial fitness in this setting. These findings suggest that IL-33

signaling is required for CNS tolerance to the inflammatory response triggered by neuroinvasive flavivirus infection.

## Results

### IL-33/ST2 signaling promotes host survival upon neuroinvasive flavivirus infection

We sought to assess the role of IL-33 and its receptor ST2 in controlling the CNS pathogenesis of flavivirus infection. In order to isolate roles for IL-33 and ST2 within the CNS, we used an established model of intracranial infection with an attenuated form of WNV (WNV-E218A), a model in which most wild-type mice survive infection but display CNS pathology and sequelae [17,22] (**S1A Fig**). To study the role of the IL-33/ST2 pathway in these infection models, we infected $Il1rl1^{-/-}$, $Il33^{-/-}$, and wild type congenic animals with WNV-E218, then monitored their survival. While ~75% of wild type animals survived infection, both $Il1rl1^{-/-}$ and $Il33^{-/-}$ mice were significantly more susceptible to WNV-E218A infection, displaying reduced survival and enhanced weight loss (**Figs 1A and S1B**). We reasoned that this increase in susceptibility could be due to a failure to control viral replication within the CNS. To test this, we titered virus from the cortex, hippocampus, cerebellum, brainstem, and peripheral organs of $Il1rl1^{-/-}$, $Il33^{-/-}$, and B6/J infected mice at 7 days post-infection (DPI), a timepoint corresponding to peak disease [17]. While no virus was detected in the periphery of these animals, only the cortex exhibited varying degrees of infectious virus titer (**Fig 1B**). Surprisingly, $Il33^{-/-}$ animals displayed lower levels of viral titers in this brain region compared to controls while $Il1rl1^{-/-}$ titers were higher than WT. No differences in viral titer were seen in other CNS regions.

To assess broader roles for the IL-33/ST2 axis in flavivirus control, we tested the effects of IL-33 and ST2 deletion with a wild-type isolate of the related flavivirus Zika virus, termed ZIKV-Dakar. While intracranial infection of congenic WT animals yielded ~90% survival, $Il1rl1^{-/-}$ mice were significantly more susceptible to infection with survivorship at ~70%. $Il33^{-/-}$ mice had a further decrease in survivorship to both congenic controls and $Il1rl1^{-/-}$ mice at ~30% with both genotypes showing increased weight loss (**Figs 1C and S1C**). Upon examination of viral titers from the tissues of these animals we observed that, similarly to WNV-E218A infection, no virus was detected in peripheral tissues (**Fig 1D**). Within the CNS only the brainstem showed differences in viral titer upon loss of IL-33/ST2 signaling upon ZIKV-Dakar infection, with $Il1rl1^{-/-}$ mice showing lower levels of infectious virus than either WT or $Il33^{-/-}$ animals (**Fig 1D**). WNV-E218A titers post-peak infection (10 DPI) in $Il1rl1^{-/-}$ and congenic controls only showed elevated titers in the cerebellum, with all other brain regions having cleared virus (**S1D Fig**). We also assessed the role of IL-33/ST2 signaling in models of peripheral flavivirus infection, and observed increased mortality and weight loss in $Il1rl1^{-/-}$ mice (**S1E and S1F Fig**). Further, titering infected tissues of these mice showed no statistically significant differences in CNS or peripheral organ viral titer (**S1G Fig**). Since the cerebellum of $Il1rl1^{-/-}$ mice showed an increase in viral titer (**S1D Fig**), and to assess the effect of IL-33 on CNS viral restriction over time we intracranially infected $Il33^{-/-}$ and congenic controls with WNV-E218A, and collected CNS and peripheral tissues at early or late timepoints, corresponding to day 4 (**Fig 1E**) and 14 (**Fig 1F**) post-infection, respectively. Four days post-infection, we observed limited viral titers across brain regions, while by 14 days post-infection virus had been uniformly cleared, with only very modest viral titers observed in the cortex. At neither timepoint were significant differences in viral titer observed between wild-type and $Il33^{-/-}$ mice. Collectively these results suggest that the IL-33/ST2 signaling pathway is important for host survival upon neuroinvasive WNV infection, but that this activity is independent of a role in controlling viral replication or dissemination.

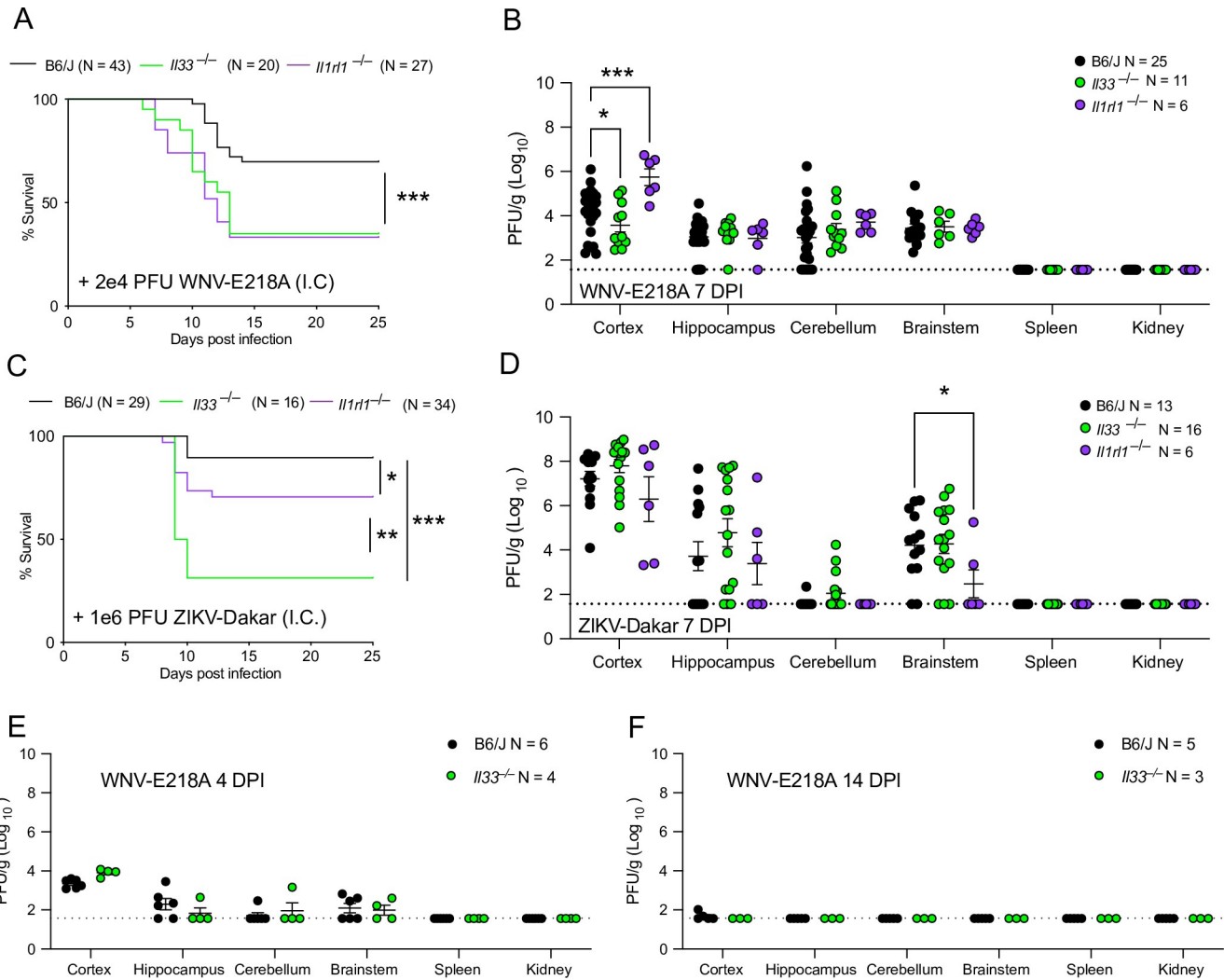

**Fig 1. The IL-33 pathway promotes survival upon neuroinvasive flavivirus infection, independent of viral control.** (A) Survival analysis in *Il33*<sup></sup>−/−,
*Il1rl1*−/−, and B6/J mice following intracranial infection with 2e4 PFU WNV-E218A. (B) WNV-E218A Viral titers from indicated CNS components and
tissues of mice 7 days post-infection via plaque assay on BHK cells. (C) Survival analysis in *Il33*−/−, *Il1rl1*−/−, and B6/J mice following intracranial infection of
1e6 PFU ZIKV-Dakar. (D) ZIKV-Dakar Viral titers from indicated CNS components and tissues of mice 7 days post-infection via plaque assay on Vero
cells. (E) WNV-E218A Viral titers from indicated CNS components and tissues of mice 4 days post-infection via plaque assay on BHK cells. (F)
WNV-E218A Viral titers from indicated CNS components and tissues of mice 14 days post-infection via plaque assay on BHK cells. Data are representative
of 3 pooled independent experiments (error bars, SEM). * p < 0.05, ** p < 0.01, *** p < 0.001 (A, C) (Gehan-Breslow-Wilcoxon test) (B, D, E) (2-way
ANOVA with Holm-Sidak Multiple comparisons).

## Oligodendrocyte-derived IL-33 promotes survival upon neuroinvasive WNV infection

Since our studies of knockout mice suggested a role for IL-33 in promoting survival to neuroinvasive flavivirus infection, we next sought to assess the CNS cell source of IL-33 in this setting. We first analyzed the expression and localization of IL-33 in the uninfected CNS and found that IL-33 was exclusively expressed in either cells of the oligodendrocyte lineage (Olig2 +) or astrocytes (GFAP+) in concordance with prior studies[8,12] **(Fig 2A).** Since IL-33 signaling is associated with its release from dying cells, the loss of IL-33<sup>+</sup> cells has been used as a surrogate for IL-33 activity [8,12]. To examine the release of IL-33 by Olig2 or GFAP-expressing cells, we intracranially infected C57Bl6/J mice with WNV-E218A and sacrificed animals

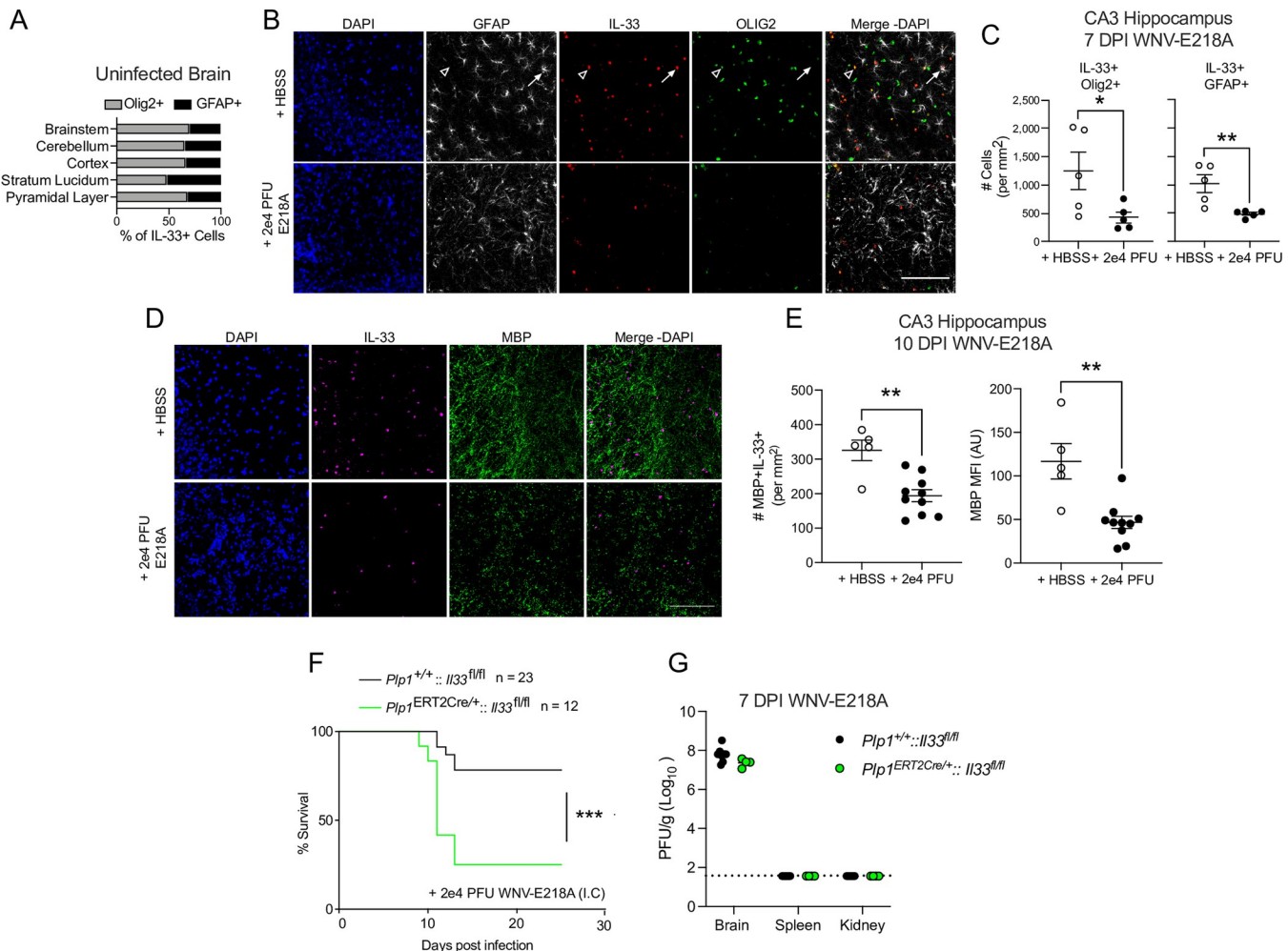

**Fig 2. Oligodendrocyte-derived IL-33 promotes survival to neurotropic flavivirus infection.** (A) Ratios of IL-33+ cells in indicated brain regions in adult B6/J mice with HBSS injection. (B) Representative immunofluorescent images of IL-33+ cell loss in B6/J Hippocampal CA3 7 days post intracranial infection with WNV-E218A (20x magnification, scale bars, 100 microns). Arrowheads indicate IL-33+Olig2+ and arrows IL-33+GFAP+ cells. (C) Quantification of IL-33 positivity in Olig2+ and GFAP+ in B6/J Hippocampal CA3 7 days post intracranial infection with WNV-E218A. (D) Representative images of MBP+IL-33 + cell loss in hippocampal CA3 10 days post intracranial infection with WNV-E218A (20x magnification, scale bars, 100 microns). (E) Quantification of MBP +IL-33+ cell number and MPB mean fluorescent intensity in Hippocampal CA3 10 days post intracranial infection with WNV-E218A. (F) Survival analysis of mice with oligodendrocyte-specific deletion of IL-33 following intracranial infection of 2e4 PFU WNV-E218A. (G) WNV-E218A Viral titers from indicated tissues of mice 7 days post-infection via plaque assay on BHK cells. (A-E) Data are representative of two independent experiments. (F) Data are representative of 3 pooled independent experiments (error bars, SEM). * $p < 0.05$, ** $p < 0.01$, *** $p < 0.001$. (C and E) two-tailed student's T test (F) (Gehan-Breslow-Wilcoxon test).

during peak CNS disease at 7 DPI, preserving their brains for IHC analysis. Examining hippocampal sub region CA3, an area receiving myelination from septal GABAergic neurons[23,24] and known for synaptic stripping and inflammation during WNV-E218A infection, we observed a loss of both IL-33+ astrocytes (GFAP+) and IL-33+ oligodendrocyte-lineage cells (Olig2+) 7 days post-infection (**Fig 2B and 2C**).

Since IL-33 release is classically associated with cell death we next quantified the number of both astrocytes and oligodendrocytes post-infection. While total GFAP+ astrocyte numbers were stable (**S2A Fig**), IL-33+ mature-myelinating oligodendrocyte cell numbers (MBP+IL-33 +) and overall MBP staining were decreased post-infection (**Fig 2D and 2E**). Given the loss of MBP+IL-33+ oligodendrocytes in regions of known synaptic stripping as well as previous

studies that have shown oligodendrocyte cell death upon flavivirus infection *in vitro* and *in vivo*[25–27], we hypothesized that oligodendrocytes represented a probable source of released IL-33 following neuroinvasive WNV infection. To test this, we crossed *Plp1*$^{ERT2Cre/+}$ mice with *Il33*$^{fl/fl}$ animals to generate mice lacking IL-33 in oligodendrocytes. We first confirmed that tamoxifen treatment induced the excision of IL-33 from Olig2+ cells without impacting total Olig2+ cell number (**S2B and S2C Fig**). We next intracranially infected tamoxifen-treated *Plp1*$^{ERT2Cre/+}$::*Il33*$^{fl/fl}$ mice, as well as Cre-negative littermates, with WNV-E218A. Strikingly, Cre+ animals exhibited a phenotype closely matching that of global IL-33 deficient animals, with compromised survival (**Fig 2F**) but no differences in viral titer between genotypes at 7 DPI (**Fig 2G**). Collectively, these data indicate that oligodendrocyte loss is widespread following CNS WNV infection, and that oligodendrocyte-derived IL-33 promotes host survival independently of viral restriction.

## The IL-33/ST2 signaling axis promotes survival independent of adaptive immunity during neuroinvasive flavivirus infection

To assess whether loss of virologic control may occur at sites of adaptive immune activation in the absence of IL-33 signaling we assessed viral titer in the deep cervical lymph nodes (DCLNs) of *Il33*$^{-/-}$ mice and congenic controls at early (4 DPI) and late (14 DPI) stages post WNV-E218A intracranial infection. Though the DCLNs have been implicated in T cell activation following CNS infection[28,29], we observed no titerable virus at these timepoints (**Fig 3A**).

Despite this, the IL-33/ST2 signaling pathway has well-defined roles in T cell activation during viral infection, with CD8+ [30], CD4+ [31], and Treg cells [32] implicated. Notably, these three adaptive immune cell types are all critical for WNV host defense [33,34]. We therefore wondered if the survival defect observed in *Il33*$^{-/-}$ or *Il1rl1*$^{-/-}$ mice upon neuroinvasive flavivirus infection was due to altered T cell recruitment or activation. To examine this, we infected both WT and *Il33*$^{-/-}$ mice intracranially with WNV-E218A or ZIKV-Dakar and sacrificed each at 7 days post-infection, removing and processing the brain parenchyma for examination of T cell responses by flow cytometry. Remarkably, there was no change in the number or expression of activation markers (CD44 and CD69) on adaptive effector cells in response to WNV-E218A in *Il33*$^{-/-}$ animals (**Fig 3B**). While there was a modest increase in overall T cell number with ZIKV-Dakar infection in *Il33*$^{-/-}$ mice, no differences in Tconv, Treg, or CD8 T cells were seen in *Il33*$^{-/-}$ animals with ZIKV-Dakar infection, with MFIs of CD44 and CD69 unchanged in Tconv and CD8 T cells (**Fig 3C**). T cell responses remained unchanged in *Il33*$^{-/-}$ mice 14 DPI with WNV-E218A with no difference in either cell number or expression of effector markers (**Fig 3D**). Given the essential role of CD8 T cells in CNS protection during WT WNV peripheral infection [35], we sought to functionally test their role by depleting CD8 T cells from WT animals and infecting both control and CD8 T cell-depleted mice with WNV-E218A to monitor survival (**S3A Fig**). Despite robust CD8 T cell depletion, especially among NS4b+ CD8 T cells specific to WNV (**S3B Fig**)**,** no change in survival or weight loss was observed, in contrast to effects of CD8 depletion following peripheral WNV infection [35] (**S3C and S3D Fig**).

To further assess a role for IL-33/ST2 signaling in adaptive immunity, we sought to define which leukocyte subsets express ST2 during infection. To do this, we analyzed both mock and WNV-E218A infected WT mice by flow cytometry at 7 DPI to investigate ST2 expression on CNS immune cells (**S3E Fig**). While HBSS-injected animals showed highest ST2 expression on CD8+ and CD4+ T cells (**Fig 3E**), during infection we observed the highest proportion of ST2+ cells among Tregs (~8%), followed by Tconv (~2%), and CD45+Non-LymphoidCD11b-

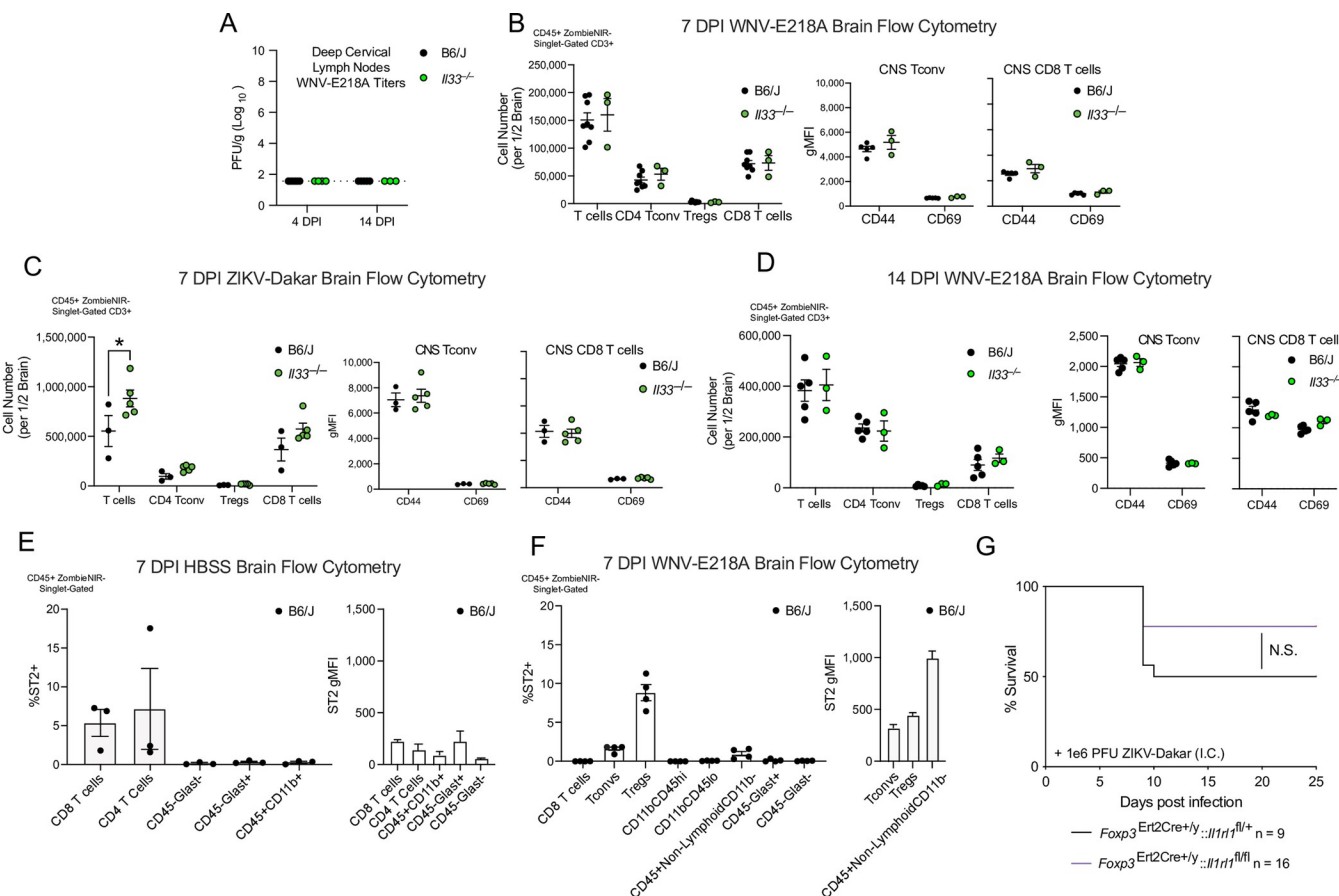

**Fig 3. IL-33/ST2 signaling on adaptive immune cells does not impact host survival following intracranial flavivirus infection. (A)** WNV-E218A Viral titers from DCLNs of mice 4- and 14-days post-infection via plaque assay on BHK cells. (B) Cell number and CD44/CD69 gMFI of indicated adaptive immune cell types per ½ brain by FACs 7 days post intracranial infection with WNV-E218A. (C) Cell number and CD44/CD69 gMFI of indicated adaptive immune cell types per ½ brain by FACs 7 days post intracranial infection with ZIKV-Dakar. (D) Cell number and CD44/CD69 gMFI of indicated adaptive immune cell types per ½ brain by FACs 14 days post intracranial infection with WNV-E218A. (E) Percent ST2+ and ST2 gMFI of indicated immune cell types 7 days post vehicle injection. (F) Percent ST2+ and ST2 gMFI of immune cells 7 days post intracranial infection with WNV-E218A. (G) Survival analysis of mice with Treg deletion of ST2 following intracranial ZIKV-Dakar infection. Data are representative of 2 pooled independent experiments (error bars, SEM). * p < 0.05, ** p < 0.01, *** p < 0.001 (A, B) (2-way ANOVA with Holm-Sidak Multiple comparisons) (C, D) (One-way ANOVA with Holm-Sidak Multiple comparisons). (E) (Gehan-Breslow-Wilcoxon test). Data are representative of 2 pooled independent experiments (error bars, SEM). * p < 0.05, ** p < 0.01, *** p < 0.001. (B, D, G, J, H, L) (2-way ANOVA with Holm-Sidak Multiple comparisons) (C, K) (Gehan-Breslow-Wilcoxon test) (I) (Student's two-tailed t Test).

cells (~1%); the latter likely representing meningeal innate lymphoid type 2 cells (ILC2s) which are known to express ST2 [36] and exhibited the highest levels of ST2 by MFI (**Figs 3F and S3F**). As Tregs are known to be critical for host survival following WNV infection and exhibited the highest proportion of ST2 positivity by any of the examined leukocytes (**Figs 3F and S3F**) we tested whether lack of IL-33/ST2 signaling on Treg cells contributed to the observed survival defects following neuroinvasive flavivirus infection. To assess this, we crossed *Foxp3*[ERT2Cre+/y] mice with *Il1rl1*[fl/fl] animals, to delete ST2 on FOXP3-expressing Treg cells. After confirming that ST2 expression was deleted on both splenic and tissue Tregs prior to infection (**S3G Fig**) we intracranially infected mice with both intact (*Foxp3*[Ert2Cre+/y]:: *Il1rl1*[fl/+]) and deleted (*Foxp3*[Ert2Cre+/y]:: *Il1rl1*[fl/fl]) Treg ST2 with ZIKV-Dakar. Despite effective ST2 deletion on Tregs we observed no difference in survival (**Fig 3G**) or weight loss (**S3H Fig**) between groups, indicating that Treg expression of ST2 is dispensable for CNS flavivirus defense. As an alternative approach to assess a role for Tregs in survival following

neuroinvasive WNV infection, we boosted the number and reactivity of Tregs via administration of IL-2 complexes [37] following CNS infection of WNV-E218A. While both the number of Tregs and expression of effector proteins CD25 and GITR were increased in circulation 9 DPI (**S3I and S3J Fig**)**,** no effect of survival or weight loss to WNV-E218A infection was observed in response to IL-2 complex administration (**S3K and S3L Fig**). Collectively, these data suggest that while ST2 expression is prominent on adaptive immune cells during peak CNS disease, signaling via the IL-33/ST2 axis on adaptive immune cells is dispensable for direct CNS immune responses to flaviviruses.

## IL-33 signaling is required for normal microglial activation following neuroinvasive WNV infection

We next sought to identify an effector cell population for the IL-33/ST2 axis during CNS flavivirus infection. Given our negative findings regarding roles for lymphocytes in this process, as well as recent work highlighting a role for IL-33 signaling in microglial activation [6,7,38] we sought next to characterize the role of IL-33 on microglial activation following intracranial WNV-E218A infection. Due to the difficulty of discriminating microglia versus infiltrating monocyte-derived macrophages we assessed the stability of the microglial-specific marker TMEM119[39] post-infection. While TMEM119 was expressed on nearly all cells expressing the pan-macrophage marker Iba1[40] 10 days post vehicle injection, only ~20% of Iba1+ cells expressed TMEM119 10 days post WNV-E218A infection (**Fig 4A**) indicating either a loss of TMEM119 expression by resident microglia or an influx of TMEM119- monocyte-derived macrophages.

The marker phopsho-p38 has been shown to be acutely expressed by ST2+ adaptive immune cells following IL-33 exposure both *in vivo* and *in vitro* [30,31,41]. To examine expression of phospho-p38 in activated brain macrophages post WNV-E218A infection we stained 10 DPI CNS tissue for phospho-p38, the pan-macrophage marker Iba1, and the phagolysosomal marker CD68, a marker expressed highly on brain macrophages during WNV-induced synaptic stripping [17]. Comparing CA3 tissue from $Il1r1l^{-/-}$ and congenic control mice we observed reduced brain macrophage activation quantified both by Iba1 MFI and Iba1+CD68+phospho-p38+ brain macrophage number 10 DPI (**Fig 4B and 4C**).

These data suggested that IL-33/ST2 signaling may drive microglia activation following WNV infection. We next examined brain macrophage activation more broadly by examining Iba1 MFI in regions of known WNV infectivity in both WT and $Il1rl1^{-/-}$ mice 7 DPI with WNV-E218A. Iba1 MFI was reduced in CA3, deep cerebellar nuclei (DCN), and cortex of $Il1rl1^{-/-}$ mice compared to congenic controls (**Fig 4D and 4E**). Importantly, mock-infected $Il1rl1^{-/-}$ mice did not show decreased Iba1+ cell number, indicating no baseline effect of IL-33 on promoting microglia number (**S4A and S4B Fig**) Despite its reported role in synaptic stripping by activated microglia in diverse CNS pathologies [42], complement deposition of C1q protein at 7DPI was not affected by the absence of IL-33 during infection (**S4C and S4D Fig**)**.**

To directly test whether the loss of IL-33/ST2 signaling on microglia, and resulting impairment in activation, was responsible for the increased susceptibility to WNV infection we previously observed in IL-33- and ST2-deficient mice, we crossed $Cx3cr1^{Ert2Cre}$ mice with $Il1rl1^{fl/fl}$ animals. In resulting $Cx3cr1^{Ert2Cre/+}$:: $Il1rl1^{fl/fl}$ animals, tamoxifen administration causes ST2 excision in all CX3CR1-expressing cells, including tissue resident macrophages in both the CNS and periphery. However, we used a previously-defined strategy of administering tamoxifen, then resting mice for one month to allow turnover of peripheral, but not brain-resident, macrophage population [43]. While activated T cells have also been shown to express CX3CR1, no T cell labeling via this strategy has been observed [43,44]. We first confirmed

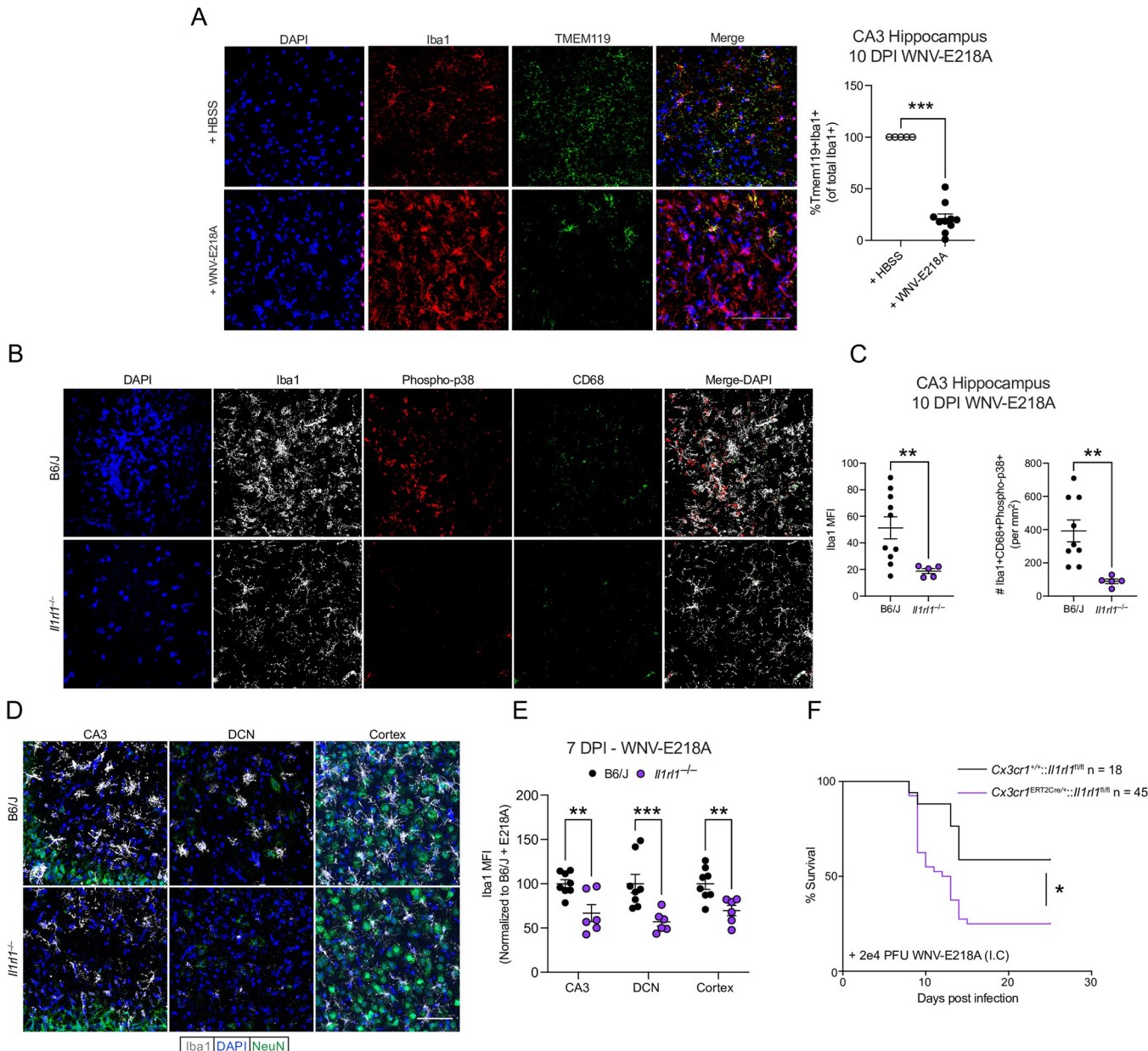

**Fig 4. IL-33/ST2 signaling is necessary for microglial activation and host defense following CNS flavivirus infection.** (A) Representative immunofluorescent images and quantification of microglial TMEM119 loss in hippocampal CA3 post WNV-E218A infection (20x magnification, scale bars, 100 microns). (B) Representative images and (C) Quantification of Iba1 MFI and Iba1+phospho-p38+CD68+ brain macrophage number post intracranial WNV-E218A infection. (D) Representative immunofluorescent images of Iba1+ brain macrophages in B6/J and $Il1rl1^{-/-}$ mice 7 days post-infection with WNV-E218A (20x magnification, scale bars, 100 microns). (E) Quantification of Iba1 by MFI in indicated brain regions of B6/J and $Il1rl1^{-/-}$ mice 7 days post-infection with WNV-E218A. (F) Survival analysis of mice with microglial-specific deletion of ST2 following intracranial infection with WNV-E218A. Data are representative of 2 and 3 (E) pooled independent experiments (error bars, SEM). * $p < 0.05$, ** $p < 0.01$, *** $p < 0.001$. (B, D) (2-way ANOVA with Holm-Sidak Multiple comparisons and Pearson's Correlation) (E) (Gehan-Breslow-Wilcoxon test).

microglial ST2 excision in naïve mice one-month post tamoxifen treatment by MAC sorting whole brain homogenate with CD11b beads and subsequent RT-PCR. After first confirming purity of our samples by flow cytometry (**S4E and S4F Fig**), we observed downregulation of the ST2 receptor only in $Cx3cr1^{Ert2Cre/+}$:: $Il1rl1^{fl/fl}$ sorted microglia by RT-PCR (**S4G Fig**). Next, we infected tamoxifen-treated $Cx3cr1^{Ert2Cre/+}$:: $Il1rl1^{fl/fl}$ and $Cx3cr1^{+/+}$:: $Il1rl1^{fl/fl}$ animals

one-month post-tamoxifen treatment intracranially with WNV-E218A and monitored their survival. While animals with intact ST2 signaling on microglia survived at rates comparable to WT animals, $Cx3cr1^{ERT2Cre/+}$:: $Il1rl1^{fl/fl}$ mice displayed significantly increased mortality (**Fig 4F**) and weight loss (**S4H Fig**), comparable to that observed when this infection model was applied to mice globally lacking ST2 or IL-33 (**Fig 1A**). A recent study identified ST2 expression on peripheral sensory neurons in the context of sterile skin injury [45]. To test whether neuronal ST2 expression contributed to our phenotypes we crossed $Camk2a^{Cre/+}$ mice with $Il1rl1^{fl/fl}$ mice generating mice with intact ($Camk2a^{+/+}$:: $Il1rl1^{fl/fl}$) and deficient ($Camk2a^{Cre/+}$: $Il1rl1^{fl/fl}$) neuronal ST2 signaling. Upon WNV-E218A infection no change in survival (**S4I Fig**) or weight loss (**S4J Fig**) was seen. Together, these findings support the idea that ST2 expression on microglia is responsible for both macrophage activation and host survival in response to WNV infection.

## Brain macrophages from $Il33^{-/-}$ mice exhibit enhanced inflammatory and stress responses post-infection

Due to a loss of microglial markers post-infection (**Fig 4A**) and a requirement for ST2 expression on microglia to support host survival (**Fig 4F**) we sought to identify the transcriptional programs initiated by IL-33 on brain macrophages in total post-infection. We first isolated hippocampal brain macrophages (including both microglia and recruited peripheral monocyte-derived macrophages) using F4/80 mediated MACS from WT and $Il33^{-/-}$ mice 7 days after intracranial WNV-E218A infection and processed them for RNA sequencing. (**Fig 5A**). Before sequencing, MAC-sorted sample purity was determined by flow cytometry (**S5A–S5C Fig**). Following RNAseq analysis, we first performed principal component analysis and observed clustering of samples by both genotype and infection status (**Fig 5B**). Analysis of RNAseq datasets revealed 251 significantly differentially expressed genes when comparing B6/J and IL-33 deficient macrophages during infection (**Fig 5C**). Additionally, and consistent with previous reports, we observed a number of differentially expressed genes in vehicle-treated IL-33-deficient animals relative to wild-type controls, consistent with homeostatic roles for IL-33 [38] (**S5D and S5E Fig**). We next performed Gene Set Enrichment Analysis (GSEA) using Hallmark gene sets that may be differentially expressed during infection. While the top gene sets enriched in IL-33-deficient cells relative to wild-type during infection (**Fig 5D**) included predictable dysregulation of inflammatory processes (**S5F Fig**), we were surprised to observe a number of cell metabolism and stress gene sets enriched including: Fatty Acid Metabolism (**S5G Fig**), Oxidative Phosphorylation (**S5H Fig**), Hypoxia (**S5I Fig**) and Apoptosis (**S5J Fig**). Despite evidence of loss of the microglial marker TMEM119 at the protein level (**Fig 4A**), no changes to microglial homeostatic signature genes associated with genotype aside from $C1qb$ were detected in vehicle or virus-infected brain macrophages (**Fig 5E**). Still, this does not discount the possibility of bone-marrow derived macrophages in our samples. Collectively, these data point to a dysregulation of brain macrophage activation following WNV-E218A infection in $Il33^{-/-}$ mice typified by altered metabolic state and upregulation of cell stress pathways, including apoptosis.

## IL-33 functions as a microglial survival factor limiting CNS pathology following WNV infection

Previous studies have shown a role for IL-33 in the metabolic support of macrophage survival [5,38]. Since we observed a lack of macrophage activation and hypertrophy in ST2-deficient mice (**Fig 4B–4E**), as well as upregulated cell stress pathways in brain macrophages isolated from $Il33^{-/-}$ mice (**Fig 5D**), we hypothesized that brain macrophage viability may be impacted

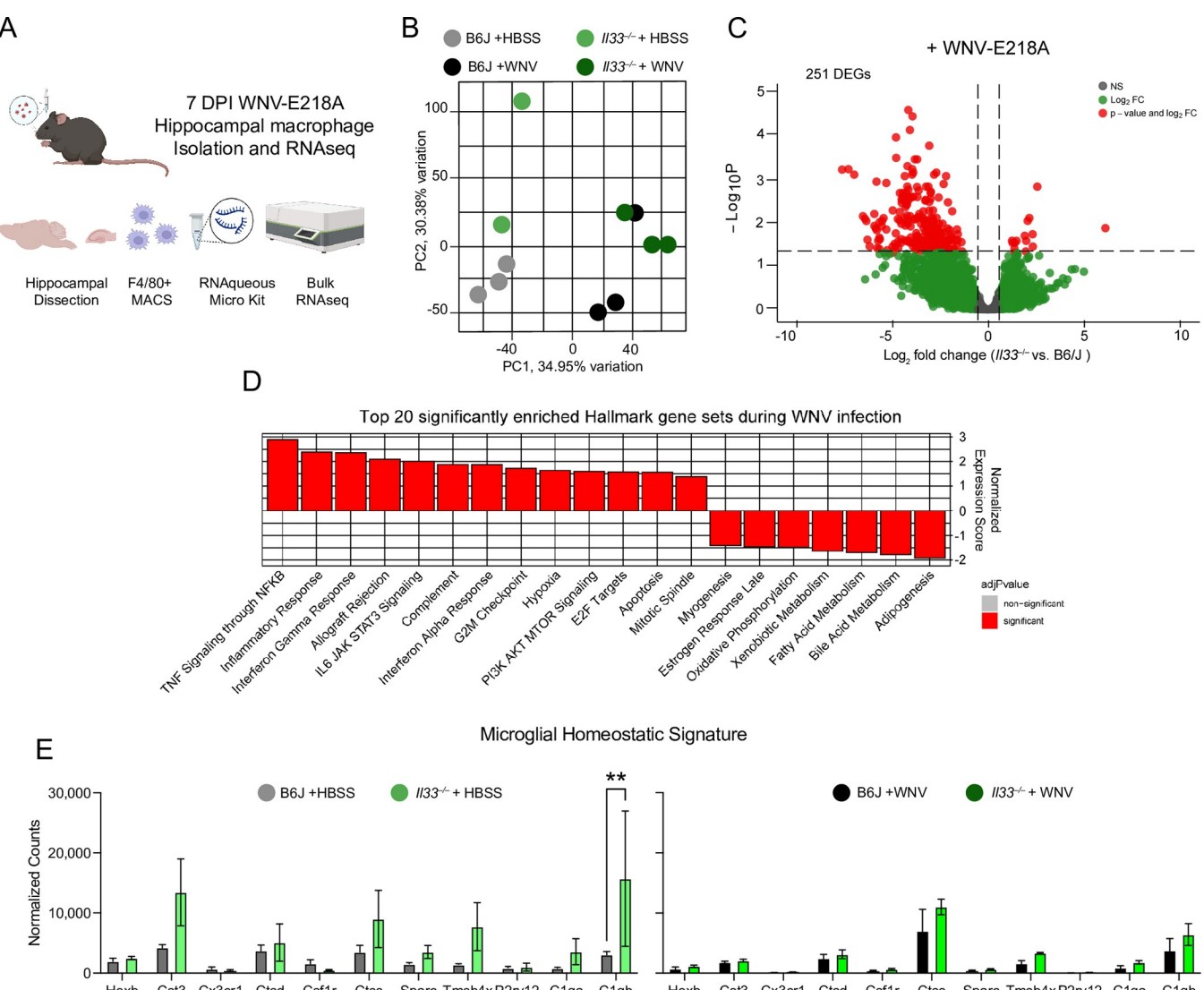

**Fig 5. *Il33⁻/⁻* hippocampal macrophages exhibit enhanced inflammatory and cell stress responses acutely during CNS WNV infection.** (A) Scheme of hippocampal macrophage isolation and RNAsequencing following intracranial WNV-E218A infection of B6/J and *Il33⁻/⁻* mice. Created using Biorender.com and exported via an academic subscription belonging to AO. (B) Principal Component Analysis of RNAseq samples. (C) Volcano plot showing differential gene expression in *Il33⁻/⁻* vs. B6/J hippocampal macrophages post-infection with WNV-E218A. (D) Top 20 Hallmark gene sets enriched in *Il33⁻/⁻* relative to B6/J hippocampal macrophages post-infection with WNV-E218A. (E) Microglial homeostatic signature genes in hippocampal microglia isolated post-vehicle and infection with WNV-E218A. ** p < 0.01 (E) 2-way ANOVA with Holm-Sidak Multiple comparisons.

subsequent to intracranial WNV-E218A delivery in the absence of IL-33/ST2 signaling. To quantify this, we examined brain macrophage (CD11b+F4/80+) viability in *Il1rl1⁻/⁻* mice post-infection by flow cytometry. Looking 7 days post-infection with WNV-E218A we observed that in whole brain samples, macrophages, but not other infiltrating leukocyte populations exhibited higher proportions of cell death as quantified by the viability dye ZombieNIR (**Fig 6A**). To elucidate the extent of this phenotype we also analyzed *Il33⁻/⁻* mice and congenic controls at 14 DPI with WNV-E218A, and again found that only brain macrophage populations showed enhanced ZombieNIR positivity (**S6A Fig**). Examination of innate immune cell counts by flow cytometry at 14 DPI also showed decreased numbers of brain myeloid cells and brain macrophages, consistent with enhanced cell death in these populations (**S6B Fig**). To confirm

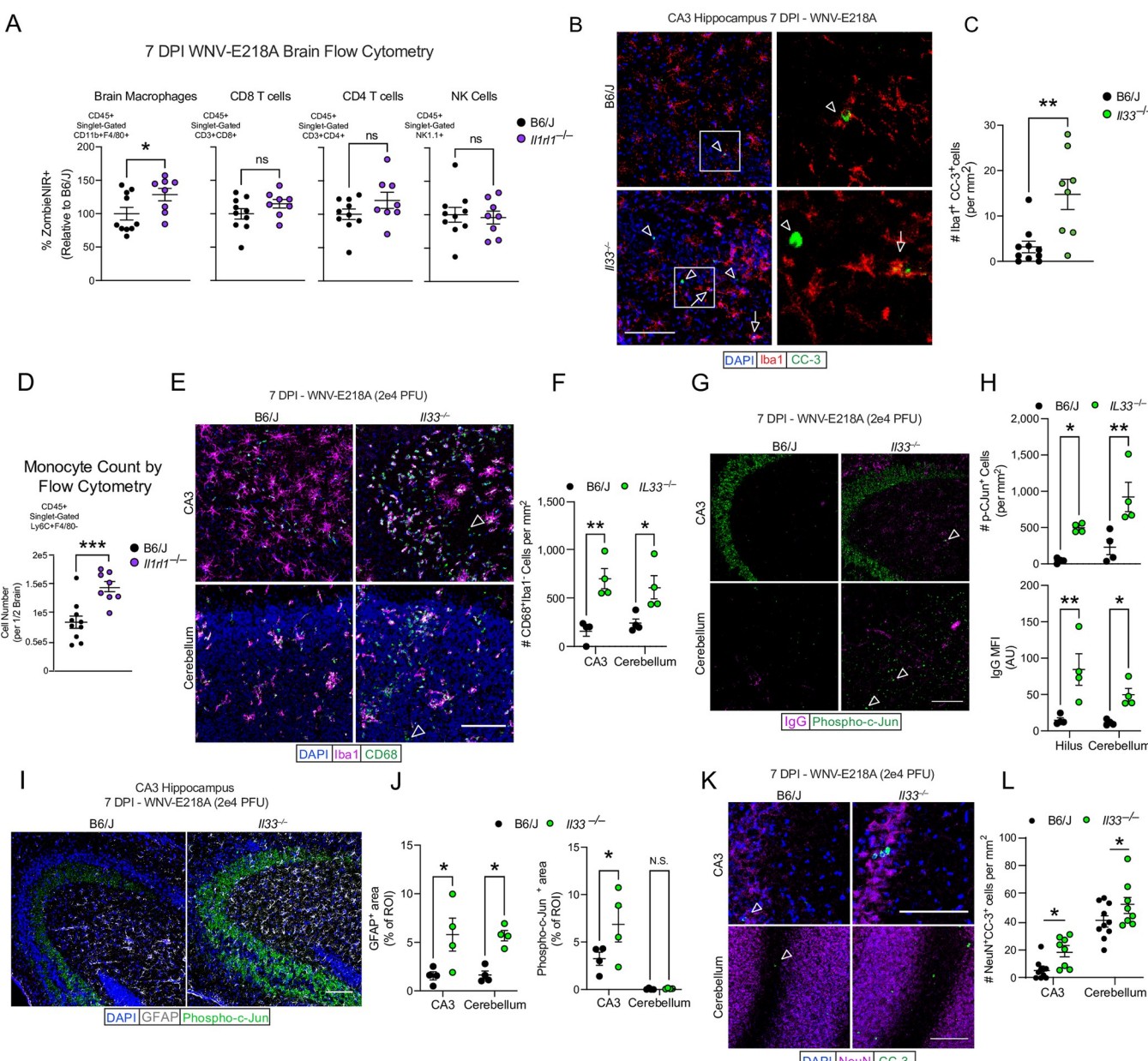

**Fig 6. Disruption of IL-33 signaling during CNS WNV infection induces brain macrophage apoptosis and neuronal pathology.** (A) %ZombieNIR+ dying cell populations of indicated type by FACs analysis of brain homogenates 7 days post-infection with WNV-E218A in B6/J and $Il1rl1^{-/-}$ mice. (B) Representative images of CC3+ CA3 hippocampal brain macrophages 7 days post-infection with WNV-E218A in B6/J and $Il33^{-/-}$ mice (20x magnification, scale bars, 100 microns). Arrowheads indicate Iba1-CC3+ while arrows indicate Iba1+CC3+ cells. (C) Quantification of CC3+ CA3 hippocampal brain macrophages. (D) Quantification of monocyte count by FACs analysis of brain homogenates in B6/J and $Il1rl1^{-/-}$ mice 7 days post WNV-E218A infection. (E) Representative images of monocyte engraftment in hippocampal CA3 and cerebellum of B6/J and $Il1rl1^{-/-}$ mice 7 days post WNV-E218A infection (20x magnification, scale bars, 100 microns). Arrowheads indicate Iba1-CD68+ cells. (F) Quantification of CD68+Iba1- monocytes in hippocampal CA3 and cerebellum. (G) Representative images of pCJun+ cells and IgG deposition in hippocampal CA3 7 days post-infection with WNV-E218A in B6/J and $Il33^{-/-}$ mice (20x magnification, scale bars, 100 microns). Arrowheads indicated pCJun+ cells. (H) Quantification of pCJun+ cells and IgG MFI. (I) Representative images of pCJun+ projections and GFAP+ cells hippocampal CA3 7 days post-infection with WNV-E218A in B6/J and $Il33^{-/-}$ mice (20x magnification, scale bars, 100 microns). (J) Quantification of GFAP+ area and pCJun+ projection area. (K) Representative images of CC3+ neurons in hippocampal CA3 and cerebellum of B6/J and $Il1rl1^{-/-}$ mice 7 days post WNV-E218A infection (20x magnification, scale bars, 100 microns). Arrowheads indicate NeuN+CC3+ cells. (L) Quantification of CC3+ neuron number in CA3 and cerebellum. Data are representative of 2 (A-F) independent and 2 (K-L) pooled independent experiments (error bars, SEM). * $p < 0.05$, ** $p < 0.01$, *** $p < 0.001$. (A, C, H) (two-tailed T test) (E, G, J, L) (Two-way ANOVA with Holm-Sidak multiple comparisons).

this finding, we next performed IHC on B6/J and $Il33^{-/-}$ mice infected with WNV-E218A at 7 days post infection, using the macrophage marker Iba1 and terminal apoptotic marker cleaved-caspase 3 (CC3). We observed increased macrophage apoptosis in the CA3 of $Il33^{-/-}$ mice with WNV-E218A infection (**Fig 6B and 6C**) and also in hippocampal CA1 7 DPI with ZIKV-Dakar (**S6C and S6D Fig**).

A key driver of pathology in neuroinvasive WNV disease is the entry of peripheral monocytes to the CNS [46–48]. Given the blunted activation and enhanced apoptosis observed in resident microglia in the absence of IL-33/ST2 signaling, we wondered if peripheral monocytes would be recruited in greater numbers in these animals to compensate for—or in response to—dysregulated microglial responses. To assess this, we examined the number of brain-engrafted monocytes in B6/J and $Il1rl1^{-/-}$ by flow cytometry following intracranial WNV-E218A infection and observed a marked increase in the number of monocytes in whole brain homogenates at 7DPI (**Fig 6D**). Enhanced monocyte engraftment post-infection was also corroborated by quantifying CD68+Iba1- cells in CA3 and cerebellum of B6/J and $Il33^{-/-}$ mice at 7DPI by immunohistochemistry of infected brain tissue (**Fig 6E and 6F**)

Given the strong phenotypes seen with both global and microglial deletion of IL-33 signaling on host and brain macrophage survival, we next sought to examine CNS pathologic processes that may underlie host demise. To examine neuronal pathology we stained hippocampal region CA3 and cerebellum in both B6/J and $Il33^{-/-}$ mice infected with WNV-E218A using antibodies against the neuronal distress marker phospho-c-Jun [49] and endogenous mouse IgG to mark blood-brain barrier (BBB) leakage. Strikingly, we saw an increase in CA3 IgG deposition and phospho-c-Jun+ cells in both areas of $Il1rl1^{-/-}$ animals (**Fig 6G and 6H**). To confirm a vascular origin of IgG we also stained vehicle and WNV-infected B6/J brain tissue with laminin and anti-mouse IgG, observing only intravascular IgG with vehicle treatment, and IgG tissue deposition only with infection (**S6E Fig**). We also observed an increased MFI for the astrocyte activation marker GFAP in CA3 and cerebellum, as well as increased phospho-c-Jun expression in mossy fibers of CA3 of $Il1rl1^{-/-}$ animals, an early phenotype of neurodegeneration observed in many models of CNS pathology [50] (**Fig 6I and 6J**). Collectively these data are consistent with a state of increased monocyte ingress and vascular leak, neuronal distress, and astrocyte hyperactivity in $Il33^{-/-}$ mice following flavivirus infection.

Lastly, we examined apoptotic neurons in hippocampal CA3 and cerebellum by quantifying cleaved-caspase 3 positive NeuN expressing cells and found increased neuronal apoptosis in $Il33^{-/-}$ mice at 7 DPI in both regions (**Fig 6K and 6L**). Together, these data suggest that IL-33/ST2 signaling is required during intracranial flavivirus infection to promote microglial viability and prevention of CNS pathology. Our findings indicate that in the absence of IL-33 signaling, flavivirus infection leads to brain macrophage apoptosis, ingress of pathogenic peripheral monocyte-derived cells to the CNS, neuronal distress, astrocyte hyperactivity, and subsequent enhanced neuronal apoptosis and host demise.

## Discussion

This study defines IL-33 as an oligodendrocyte-derived survival factor for microglia, and suggests that IL-33 is required during viral infection to promote normal microglial activation. In its absence, we observed wide-ranging defects in neuronal survival, blood-brain barrier integrity, and peripheral monocyte infiltration, culminating in reduced host survival. While the work detailed here does not define a mechanism for oligodendrocyte release of IL-33, previous studies have demonstrated profound WNV-induced death of oligodendrocytes *in vitro*, [25], from cultured explants [26], and *in vivo* during peripheral infection [27]. While these studies mainly indicated apoptotic death of oligodendrocytes, other works have shown that

oligodendrocytes express low levels of caspase 8 [51], a mediator of death-receptor induced apoptosis, and that TNF is a robust inducer of oligodendrocyte necrosis/necroptosis [52], implying that oligodendrocytes may be poised for pro-inflammatory cell death, especially during viral infection. These data, in addition to the finding that IL-33 reactivity is reduced, but not eliminated, by caspase 7 during apoptosis [53] suggests that oligodendrocytes may undergo non-apoptotic cell death to release IL-33 during neurotropic viral infection. Importantly, we also observe a reduction in GFAP+IL-33+ cells post-infection, and while our genetic studies support a role for oligodendrocyte-derived IL-33 in this setting, our work does not formally rule out a contribution of IL-33 derived from astrocytes in this model.

Our work also indicates that IL-33 signaling on adaptive immune populations does little to affect survival at least in direct intracranial infection of flaviviruses. This is despite our observation that ST2 expression was most widely seen on adaptive immune populations at either resting state or during peak infection. This may be attributable to reports that ST2 expression on various immune cells is very short-lived, as in the case of splenic CD8 T cells in LCMV infection [30], or can shift in response to inflammatory context such as its upregulation on macrophages and Natural Killer cells with decreased ST2 expression on ILC2s in models of airway smoke exposure [54]. Despite observing limited ST2 expression on microglia, ST2 deletion on microglial cells did impact survival phenotypes in a manner similar to germline knockout of ST2. Still, future studies may need to employ multiple timepoints, possibly on an hourly timescale, to detect ST2 expression by protein on microglia. We also acknowledge that the requirement for ST2 signaling on adaptive immune cells may be different in models in which flavivirus neuroinvasion occurs following peripheral infection, rather than via direct intracranial inoculation.

While our studies observed enhanced expression of inflammatory and cell stress gene sets on microglial populations in response to WNV infection, we were technically limited in our sorting of F4/80+ cells, which at 7 DPI include both microglia and engrafted monocyte-derived macrophages. Future experiments delineating resident microglia via genetic fate-mapping and crossing to appropriate knockout strains may be required, as no definitive microglial marker separating these cell types in disease states currently exists.

Notably, the effect of IL-33 on host defense was not due to a failure of viral clearance, as mice deficient in IL-33 signaling had equivalent viral titers at peak disease timepoints and those that survived acute infection eventually cleared virus from CNS tissues. Previous data have indicated a requirement for microglia in CNS WNV infection, with deletion pharmacologically [20] or genetically [22] yielding decreased host survival. However, the CNS of mice with pharmacologic microglial depletion also showed higher levels of infectious virus and a blunted adaptive immune response, contrary to what was observed here. This suggests that IL-33 signaling may drive a subset of microglial functions related to survival and disease tolerance in the context of viral infection, and that in its absence microglia may still participate in viral control. There may also be differences between genetic and pharmacological depletion of microglia, as WNV-E218A infection of microglia-deficient $Il34^{-/-}$ mice revealed decreased neuronal and host survival with no change to CNS viral titer [22], similar to our data.

While the presence of brain macrophages within the virally-infected CNS is clearly beneficial for host and neuronal survival, how are these cells providing their neuroprotective effect independent of effects on viral titer? The provision of secreted factors to neurons directly or through supporting cells is one obvious possibility. Both TGF-β and IGF-1 have been shown to be secreted by microglia during spinal cord injury and observed to promote neuronal survival at the injury site [55]. The anti-inflammatory cytokine IL-10, which is secreted by microglia in response to bacterial infection [56] may also be responsible for tuning the CNS inflammatory milieu and ensuring neuroprotection. Another possibility is microglial secretion

of brain-derived neurotrophic factor (BDNF) which is elicited by microglia during models of nerve injury and alters neuronal activity [57]. Microglial neuroprotection may also be mediated by synaptic stripping, preventing aberrant neuronal activity at sites of infection from "spillover" to otherwise healthy neuronal tissue.

IL-33 has recently emerged as a critical factor mediating macrophage survival in multiple settings. For example, IL-33 signaling was shown to promote survival in red pulp macrophage development [5] as well as altering microglial metabolic functioning [38]. Our findings join other recent studies implicating IL-33 signaling on microglia as essential to tune CNS function during development [6] and in adulthood [7]. In implicating IL-33 as a promoter of microglia-mediated disease tolerance (rather than a driver of viral clearance [58,59]) our findings highlight the pleotropic functions of microglia in both homeostasis and disease. Our work also suggests that potentiating IL-33 signaling in the CNS could ameliorate pathology and sequelae associated with neurotropic flavivirus infection.

## Materials and methods

### Ethics statement

Studies were carried out under the supervision and approval of the Institutional Animal Care and Use Committee (IACUC) of the University of Washington, protocol number 4298–01 (PI: AO).

### Mice

B6/J (000664), $Il33^{fl/fl}$ (030619) $Cx3cr1^{ERT2Cre/\ ERT2Cre}$ (020940), $Plp1^{ERT2Cre/ERT2Cre}$ (005975), Aldh1l1$^{ERT2cre/\ ERT2cre}$ (029655), and $Camk2a^{Cre/Cre}$ (005359) mice were obtained from Jackson Laboratories. C57BL/6J (B6/J) controls were either obtained commercially (Jackson Laboratories) or, for control animals in experiments using transgenic mouse strains, WT mice were bred in-house. $Il33^{-/-}$ (Originally from Dirk Smith at Amgen) and Foxp3$^{ERT2Cre/y}$ (PMID: 18387831):: $Il1rl1^{fl/fl}$ mice were obtained from the Ziegler lab (Benaroya Research Institute). $Il1rl1^{-/-}$ (ST2$^{-/-}$) (PMID: 10727469) mice were provided by the Von Moltke lab (University of Washington). Mice in this study were bred and housed under specific-pathogen free conditions at the University of Washington.

### Tamoxifen and in vivo injections

To induce Cre activity in Cre-ERT2 mice, 100 uL of Tamoxifen (Sigma) at 20 mg/mL dissolved in corn oil (Sigma) was injected every day for 5 consecutive days with mice analyzed one month later. For CD8 T cell depletion mice were injected on days -1, 2, 5, and 8 post-infection with 200 ug of αCD8 (Clone 2.43 BioXcell) or control IgG antibody in 200 uL PBS (PMID: 30143586). Depletion was confirmed by obtaining blood from the retro orbital sinus 6DPI. To induce Treg expansion (PMID: 30291203) the JES6-1 αIL-2 antibody was mixed at a 2:1 molar ratio with IL-2 for 30 minutes at 37˚C in PBS. Mice were injected daily with a complex of 1 ug IL-2 (Peprotech) and 5 ug JES6-1 (BioXcell) intraperitoneally on days 2–4 post-infection in a total volume of 200 uL. Expansion was confirmed by obtaining blood from the retro orbital sinus at 9DPI.

### Viruses

WNV-Tx was provided by Mike Gale of the University of Washington. WNV-E218A and ZIKV-Dakar strains utilized for intracranial infections were obtained from Robyn Klein at Washington University in St Louis. WNV-NS5-E218A contains a single point mutation in the

gene encoding $2'$-$O$-methyltransferase (PMID: 31611100). Mice were deeply anesthetized and intracranially administered 2e4 plaque-forming units (p.f.u.) of WNV-NS5-E218A or 1e6 PFU of ZIKV-Dakar as described previously (PMID: 31235930). Viruses were diluted in 20 µl of Hank's balanced salt solution (HBSS) and injected into the third ventricle of the brain with a guided 29-guage needle. Mock-infected mice were intracranially injected with 20 µl of HBSS. Footpad infections were performed by injection of 100 PFU WNV in 50 µL of HBSS into a rear footpad as previously described (PMID: 28366204). BHK21 cells were used for quantification of WNV Titers while Vero cells were used for titering ZIKV-Dakar as described (PMID: 24510289).

Infected mice were monitored daily for weight loss and presentation of clinical signs of disease, including hunched posture, ruffled fur, hindlimb weakness, and paresis. Severity of paresis was defined as follows: mild–partial loss of motor function in one hind limb; moderate–complete or nearly complete loss of motor function in one hind limb or partial loss of motor function in both hind limbs; severe–complete or nearly complete loss of motor function in both hind limbs and/or pronounced ataxia. Mice reaching a moribund state or losing more than 30% of initial body weight were euthanized as performed previously (PMID:28366204). All experiments were performed in both male and female 2-4-month-old mice, in accordance with protocols approved by the University of Washington Animal Care and Use Committee (IACUC).

## Measurement of viral burden

WNV- or ZIKV-infected mice were sacrificed and tissues were collected, weighed, and homogenized with Precellys zirconia beads in a Precellys 24 homogenizer in 500 µl of PBS, and stored at –80˚C until virus titration. Thawed samples were clarified by centrifugation ($2,000 \times g$ at 4˚C for 10 min), and then diluted serially before infection of BHK21 cells for WNV or Vero cells for ZIKV titers. Plaque assays were overlaid with low-melting point agarose, fixed 4 days later with 4% formaldehyde, and stained with crystal violet. Viral burden was expressed on a $\log_{10}$ scale as p.f.u. per gram of tissue.

## Tissue preparation and CNS leukocyte isolation

All tissues harvested from mice for subsequent immunohistochemical, MACs, qRT-PCR, protein, virologic, or flow cytometric analysis were extracted following extensive cardiac perfusion with 10 mL of sterile PBS after cutting the right atrium. For flow cytometry and MACs, leukocytes were isolated from whole brains after digestion in 0.05% collagenase A (Sigma-Aldrich) and 10 mg/ml DNase I (Sigma- Aldrich), then purified via centrifugation in 37% isotonic Percoll (Sigma-Aldrich) as described (PMID: 22729249). For MACs cells were incubated with anti-F4/80 microbeads (Miltenyi) and separated with LS columns (Miltenyi) per manufacturer's recommendations. FACs was performed on an LSRII (Becton Dickinson) with analysis completed on FlowJo (Becton Dickinson).

## Immunofluorescence and FACs antibodies

Brain tissue was fixed for 48 h in 4% PFA, followed by cryoprotection in 30% sucrose for 48 h. Brains were then frozen in optimal cutting temperature compound (Tissue-Tek) on the freezing element of a Leica CM3050 S cryostat. 40-µm sections were sliced into 24-well plates containing PBS with 0.05% sodium azide as previously described (PMID: 29941548). Slices were permeabilized and blocked with 0.25% Triton X-100 and 5% Normal Donkey Serum (Sigma) for 1 hour followed by overnight incubation in 0.5% BSA with primary antibody at 4˚ C. The following antibodies were used for immunofluorescence staining: rabbit anti-Iba1 (1:300; CP-

290; Biocare Medical), Goat anti-Iba1(1:300; Abcam; ab5076), rat anti-CD68 (1:300; FA-11; Biolegend), Goat anti-IL-33 (AF3626; R&D), Rabbit anti-Olig2 (1:500; AB 9610; Millipore Sigma), Rat-anti GFAP (1:2000; 2.2B10; Thermo Fischer Scientific), Rabbit anti-Sox9 (1:500; AB5535; EMD Millipore), Rabbit anti-NeuN (1:2000; D3S3I; Cell Signaling), Chicken anti-NeuN (1:300; 266 006; Synaptic Systems), Rabbit anti-C1q (1:300; ab182451; Abcam), Rabbit anti-Cleaved caspase three (1:500; ab9661; Cell-Signaling), Rabbit anti-Phospho-c-Jun (1:1000; D47G9; Cell Signaling), and Donkey anti-Mouse IgG (1:2000; A21203; Life Technologies). Slices were washed three times for 5 min, incubated for 2 h at room temperature with the appropriate secondary antibodies (all Donkey-derived from Thermo Fisher or Jackson Immunology using Alexafluor 488, 594, and 647; Life Technologies; 1:1,000), washed in PBS with DAPI (1:20,000; 62248; Thermo Fischer Scientific) washed again three times for 5 min and mounted with Aquamount (14-390-5; Thermo Fischer Scientific). Blinded image analysis including cell counting and MFI calculations were all performed with ImageJ (National Institutes of Health).

Antibody cocktails for FACs staining were used at 1:100 in 2 mM EDTA (15-575-020; Fischer Scientific) and 0.5% BSA (A7906-100G) or FOXP3/ Transcription Factor buffers (00-5523-00; eBioscience) when staining for FOXP3. Antibodies used included GLAST (130-123-555; Miltenyi), biotinylated ST2 (101001B; MD Bioscience), Streptavidin R-Phycoerythrin (S21338; Thermo Scientific) and PE-conjugated NS4b tetramers (Fred Hutch antibody core). Antibodies against CD3 (17A2; Fischer Scientific), CD4 (RM-4; Fischer Scientific), CD8 (53–6.7; Fischer Scientific), CD44 (IM7; Fischer Scientific), CD69 (H1.2F3; Fischer Scientific), FOXP3 (FJK-16s; Fischer Scientific), F4/80 (T45-2342; Fischer Scientific), NK1.1 (PK136; BioLegend), CD19 (1d3; Thermo Fisher), Ly6C (AL-21; Fischer Scientific), Ly6G (1A8; Fischer Scientific), GITR (DTA-1; Fischer Scientific), and CD25 (PC61; Fischer Scientific).

## Statistical analyses

Statistical analyses comparing two groups at one time point were done using a student's t-test in Prism software, v. 7.0a. One-way and two-way ANOVA were used appropriately and were indicated. The number of mice per group, test used, and p values are denoted in each Fig legend. Data was graphed using Prism software, v9 (GraphPad).

## Macrophage preparation and RNA sequencing

After MACs isolation, brain macrophages were washed in PBS and cell pellets prepared using the RNAqueous Micro kit (Ambion) per manufacturer's instruction. Total RNA was concentrated by evaporation using an Eppendorf Vacufuge system and was then added to lysis buffer from the SMART-Seq v4 Ultra Low Input RNA Kit for Sequencing (Takara). Reverse transcription was performed followed by 20 cycles of PCR amplification to generate full length amplified cDNA. Sequencing libraries were constructed using the NexteraXT DNA sample preparation kit with unique dual indexes (Illumina) to generate Illumina-compatible barcoded libraries. Libraries were pooled and quantified using a Qubit Fluorometer (Life Technologies). Sequencing of pooled libraries was carried out on a NextSeq 2000 sequencer (Illumina) with paired-end 59-base reads, using a NextSeq P3 sequencing kit (Illumina) with a target depth of 5 million reads per sample. Base calls were processed to FASTQs on BaseSpace (Illumina), and a base call quality-trimming step was applied to remove low-confidence base calls from the ends of reads. The FASTQs were aligned to the GRCm38 mouse reference genome, using the STAR aligner, and gene counts were generated using HTSeq-count. QC and metrics analysis were performed using the Picard family of tools (v1.134).

Following generation of the counts matrix analyses were performed using R (v4.2.2). The flow of analyses began with PCA analysis and visualization using the PCAtools package (v2.10). Normalization and differential gene expression testing analyses were performed using the DESeq2 package (v1.38.3) using the Wald statistical testing method applied two condition comparisons. The DESeq2 results were then used to generate ranks for gene set enrichment analysis (GSEA) using the fgsea package (v1.24.0) to test for enrichment of gene ontology terms using the mouse ontology gene sets (M5) collection downloaded from gsea-msigdb. The results from the GSEA and differential gene expression testing were visualized using a combination of the ggplot2 (v3.4.0), pheatmap (v1.0.12), and EnhancedVolcano (v1.16.0) packages. All code relevant to the data analyses in this study are available online at https://github.com/OberstLab. The software used for the R analyses are publicly available.

## Figure preparation

Schematic representations in Figs 5, S2, S3 and S5 were created using Biorender.com and exported via a paid academic subscription by AO.

## Supporting information

**S1 Fig. (related to Fig 1). The IL-33 pathway improves disease outcomes across multiple models of flavivirus infection.** (A) Survival analysis of adult B6/J mice exposed to increasing doses of intracranial WNV-E218A and WNV-Tx. (B) Weight loss of $Il33^{-/-}$, $Il1rl1^{-/-}$, and B6/J mice following 2e4 PFU WNV-E218A intracranial infection. (C) Weight loss of $Il33^{-/-}$, $Il1rl1^{-/-}$, and B6/J mice following 1e6 PFU ZIKV-Dakar intracranial infection. (D) WNV Viral titers from indicated CNS components and tissues of mice 10 days post-infection via plaque assay on BHK cells. (E) Survival analysis of $Il1rl1^{-/-}$ and B6/J mice following footpad infection with 100 PFU WNV-Tx. (F) Weight loss of $Il1rl1^{-/-}$ and B6/J mice following footpad infection with 100 PFU WNV-Tx. (G) WNV-Tx Viral titers from indicated CNS components and tissues of mice 7 days post-foot pad infection via plaque assay on BHK cells. Data are representative of 3 pooled independent experiments (error bars, SEM). * $p < 0.05$, ** $p < 0.01$, *** $p < 0.001$ (A, D) (Gehan-Breslow-Wilcoxon test) (B, C, E) (2-way ANOVA with Holm-Sidak Multiple comparisons).
(TIF)

**S2 Fig. (related to Fig 2). Deletion of IL-33 in oligodendrocytes with tamoxifen treatment of $Plp1^{ERT2Cre/+}$::$Il33^{fl/fl}$ mice.** (A) GFAP+ cell number by immunohistochemistry following intracranial infection with increasing doses of WNV-E218A. (B) Scheme for tamoxifen injections to induce IL-33 excision (created using Biorender.com and exported via an academic subscription belonging to AO) and representative images of IL-33 loss in cortical oligodendrocytes following tamoxifen treatment (20x magnification, scale bars, 50 microns). Arrowheads Indicate Olig2+IL-33+ cells. (C) Quantification of total cortical oligodendrocyte number and IL-33+ oligodendrocytes following tamoxifen injection. Data are representative of 3 pooled independent experiments (error bars, SEM). *** $p < 0.001$. (C) two-tailed student's T test.
(TIF)

**S3 Fig. (related to Fig 3). ST2 is highly expressed on adaptive immune subsets, but this expression is dispensable for host survival.** (A) Scheme for CD8 T cell depletion during intracranial WNV-E218A infection, created using Biorender.com and exported via an academic subscription belonging to AO. (B) Peripheral blood counts 6 DPI with WNV-E218A to confirm CD8 depletion. (C) Survival analysis of mice with CD8 T cell depletion and WNV-E218A intracranial infection. (D) Weight change in mice with CD8 T cell depletion and

control IgG injection following WNV-E218A intracranial infection. (E) Representative FACs plots and gating strategy for indicated cell types from brain homogenate 7 days post intracranial WNV-E218A infection. (F) Representative FACs plots of ST2 expression on indicated cell types from B6/J and $Il1rl1^{-/-}$ brain homogenate 7 days post intracranial WNV-E218A infection. (G) Quantification of ST2 excision on splenic and liver Tregs of indicated genotype one-month post tamoxifen treatment. (H) Weight change in mice with Treg ST2 deletion following intracranial infection with ZIKV-Dakar. (I) Scheme and cell number from peripheral blood of indicated immune cells following IL-2 complex treatment and WNV-E218A infection at 9 DPI. (J) gMFI of CD25 and GITR on peripheral blood Tregs at 9 DPI with WNV-E218A and IL-2 complex treatment. (K) Survival analysis of mice with control or IL-2 complex treatment following intracranial infection with WNV-E218A. (L) Weight change in mice with control or IL-2 complex treatment after intracranial infection with WNV-E218A. Data are representative of 2 pooled independent experiments (error bars, SEM). * $p < 0.05$, ** $p < 0.01$, *** $p < 0.001$. (B, D, G, J, H, L) (2-way ANOVA with Holm-Sidak Multiple comparisons) (C, K) (Gehan-Breslow-Wilcoxon test) (I) (Student's two-tailed t Test).
(TIF)

**S4 Fig. (related to Fig 4). Phenotypes of ST2 signaling on microglial activation during CNS Flavivirus infection.** (A) Representative immunofluorescent images of Iba1+ brain macrophages in B6/J and $Il1rl1^{-/-}$ mice 7 days post vehicle injection (20x magnification, scale bars, 100 microns). (B) Quantification of Iba1+ cell number in vehicle injected mice of indicated brain regions. (C) Representative immunofluorescent images of C1q deposition in hippocampal CA3 of B6/J and $Il33^{-/-}$ mice 7 days post-infection (20x magnification, scale bars, 100 microns). (D) Quantification of C1q gMFI by immunofluorescence in hippocampal CA3 of B6/J and $Il33^{-/-}$ mice 7 days post-infection. (E) Representative FACs plots of microglia from CX3CR1$^{ERT2Cre/+::}$ $Il1rl1^{fl/fl}$ 30 days post tamoxifen treatment and following CD11b enrichment by MACs. (F) Quantification of MACs enrichment efficiency. (G) Quantification of $Il1rl1$ RNA by qRT-PCR of sorted microglia 30 days post tamoxifen treatment and following CD11b enrichment by MACs. (H) Weight change in CX3CR1$^{ERT2Cre/+::}$ $Il1rl1^{fl/fl}$ mice following intracranial infection of WNV-E218A. (I) Survival analysis of mice with neuronal $Il1rl1$ deletion following intracranial infection with WNV-E218A. (J) Weight change of mice with neuronal $Il1rl1$ deletion following intracranial infection with WNV-E218A. Data are representative of 2 (A-F) independent and 2 (G-N) pooled independent experiments (error bars, SEM). * $p < 0.05$, ** $p < 0.01$, *** $p < 0.001$. (B, F, H, L) (two-tailed T test) (D, I, J, N) (Two-way ANOVA with Holm-Sidak multiple comparisons) (M) (Gehan-Breslow-Wilcoxon test).
(TIF)

**S5 Fig. (related to Fig 5). Hallmark pathways enriched in $Il33^{-/-}$ hippocampal macrophages during WNV infection.** (A) Scheme of hippocampal macrophage isolation and RNAsequencing following intracranial WNV-E218A infection of B6/J and $Il33^{-/-}$ mice, created using Biorender.com and exported via an academic subscription belonging to AO. (B) Representative FACS plots of MACs-sorted samples. (C) Quantification of sample purity by FACS post-MAC-sort. (D) Volcano plot showing enriched genes in $Il33^{-/-}$ hippocampal macrophages relative to B6/J with vehicle injection. (E) Top 20 significantly enriched Hallmark gene sets by GSEA of $Il33^{-/-}$ and B6/J hippocampal macrophages post-infection in vehicle-injected samples. (F) Heatmap of Inflammatory Response pathway genes in indicated samples. (G) Heatmap of Fatty Acid Metabolism pathway genes in indicated samples. (H) Heatmap of Oxidative Phosphorylation pathway genes in indicated samples. (I) Heatmap of Hypoxia pathway genes in indicated samples. (J) Heatmap of Apoptosis pathway genes in indicated samples. ***

p < 0.001 Two-way ANOVA with Holm-Sidak multiple comparisons.
(TIF)

**S6 Fig. (related to Fig 6) IL-33 signaling prevents brain macrophage apoptosis following intracranial ZIKV-Dakar infection.** (A) %ZombieNIR+ dying cell populations of indicated type by FACs analysis of brain homogenates 14 days post-infection with WNV-E218A in B6/J and *Il33*−/− mice. (B) Cell number of indicated innate immune cell types per ½ brain by FACs 14 days post intracranial infection with WNV-E218A. (C) Representative images of CC3+ hippocampal CA1 brain macrophages in B6/J and *Il33*−/− mice 7 days post-infection with WNV-E218A (20x magnification, scale bars, 100 microns). (D) Quantification of CC3+ hippocampal CA1 brain macrophages in B6/J and *Il33*−/− mice. (E) Representative images of perivascular IgG staining in hippocampal CA3 in 7 days post-infection WNV-E218A and HBSS-injected brains from B6/J mice (20x magnification, scale bars, 100 microns).
(TIF)

## Acknowledgments

We thank Pooja Jain and Valerie Sheehan for technical assistance. We thank the lab of Dr. Michael Gale, Jr., at the University of Washington for technical assistance and discussion. We thank the vivarium staff of the University of Washington Department of Comparative Medicine SLU 3.1 for their help and support. We thank all members of the Oberst Lab for their input and support. We thank the Von Moltke and Pepper labs of the University of Washington and the Ziegler lab of Benaroya Research Institute for mouse lines used in this study.

## Author Contributions

**Conceptualization:** Geoffrey T. Norris, Joshua M. Ames, Steven F. Ziegler, Andrew Oberst.

**Data curation:** Geoffrey T. Norris, Joshua M. Ames.

**Formal analysis:** Andrew Oberst.

**Funding acquisition:** Steven F. Ziegler, Andrew Oberst.

**Investigation:** Geoffrey T. Norris, Joshua M. Ames.

**Methodology:** Geoffrey T. Norris, Joshua M. Ames, Andrew Oberst.

**Resources:** Andrew Oberst.

**Supervision:** Steven F. Ziegler, Andrew Oberst.

**Validation:** Geoffrey T. Norris.

**Visualization:** Joshua M. Ames.

**Writing – original draft:** Geoffrey T. Norris, Andrew Oberst.

**Writing – review & editing:** Geoffrey T. Norris, Joshua M. Ames, Steven F. Ziegler, Andrew Oberst.

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
