## [Decision Letter · Decision Letter 0]

26 May 2023

Dear Dr. Oberst,

Thank you very much for submitting your manuscript "Oligodendrocyte-derived IL-33 functions as a microglial survival factor during neuroinvasive flavivirus infection" for consideration at PLOS Pathogens. As with all papers reviewed by the journal, your manuscript was reviewed by members of the editorial board and by several independent reviewers. In light of the reviews (below this email), we would like to invite the resubmission of a significantly-revised version that takes into account the reviewers' comments.

We cannot make any decision about publication until we have seen the revised manuscript and your response to the reviewers' comments. Your revised manuscript is also likely to be sent to reviewers for further evaluation.

Sincerely,

Kellie A. Jurado

Academic Editor

PLOS Pathogens

Alexander Gorbalenya

Section Editor

PLOS Pathogens

Kasturi Haldar

Editor-in-Chief

PLOS Pathogens

orcid.org/0000-0001-5065-158X

Michael Malim

Editor-in-Chief

PLOS Pathogens

orcid.org/0000-0002-7699-2064

In addition to addressing the reviewers comments, in the updated version if you could please ensure inclusion of higher quality images within Figures so that data/conclusions can better be reviewed.

Reviewer's Responses to Questions

**Part I - Summary**

Reviewer #1: Norris et al report on studies examining the role of IL-33 during intracranial infections with an attenuated strain of WNV and with ZIKV-Dakar. The attempt to identify cellular sources and targets of IL-33 within the flavivirus-infected CNS. They also utilized mice with global gene deletion of IL-33 and its receptor ST2 to evaluate overall survival, virologic control, and immune activation within various brain regions. They also used cell-specific deletion of ST2 to determine how IL-33 impacts the function and survival of myeloid and neural cells type. The authors conclude that IL-33 is expressed by oligodendrocytes and exerts prosurvival effects on microglia, limits monocyte entry and protects neurons from apoptosis in the context of WNV infection. The role of IL-33 and its receptor ST2 in the CNS during viral encephalitis has not been studied previously and the authors have excellent models to address these questions. While most of the experiments are performed to a high standard, in some (especially the critical experiments that led to the conclusions above) there are significant concerns regarding controls, statistical analyses and interpretation of data that need to be addressed. It is possible that some of the conclusions may change with more appropriate experimental design and statistical evaluation.

Reviewer #2: The manuscript from Norris, et al nicely demonstrates the importance of the IL-33 in models of neurotropic flavivirus infection. They find that IL-33 is necessary for survival and that oligodendrocytes are a key source of the alarmin. They observe fewer IL-33-expressing oligos during infection suggesting that IL-33 release from these cells types when they are lost during the disease process. The authors find that Iba-1 intensity is decreased in IL-33 knockout mice in comparison to controls. Mice lacking IL-33 signaling in microglia are more susceptible to infection than controls, suggesting that microglia sensing of IL-33 is critical for resistance to infection. The authors provide data that supports increased microglia death in the absence of IL-33 signaling. Overall, the data are compelling and the manuscript is well-written.

Reviewer #3: The manuscript presented by Norris et al. describes the role of IL-33 signaling in an in vivo neuronal viral infection. The manuscript is well-written, and the topic is relevant to the field. The experiments were conducted in a controlled manner. However, the major issue that I found is the lack of high-quality resolution figures. Since many of the results were based on immunofluorescence analysis, it is essential to include images that enable a better evaluation of the findings. Unfortunately, some of the figures have a resolution that is below the required standard to observe the results. Additionally, some figure legends are too small, making it challenging to comprehend the findings. Due to the inadequate images, I could not provide a better evaluation of this submission. Therefore, I suggest that the authors revise the manuscript and resubmit it with improved resolution figures. This revision will enable a better evaluation of the findings, making it easier to confirm the results as presented. I believe that this would greatly benefit the manuscript and contribute to its appreciation in the scientific community.

**Part II – Major Issues: Key Experiments Required for Acceptance**

Reviewer #1: Figure 1. There is a substantial difference in the survival curve of WT versus IL-33- or IL1rl1-deficient mice after infection with WNV-E218A. The KO mice appear to die earlier, suggesting a loss of virologic control by innate immune mechanisms, perhaps in the periphery. The authors should include cervical draining LNs in their virologic analyses to address this possibility, which is also highlighted by the ZIKV-Dak survival curves, in which none show this early effect. There is also a substantial difference in the survival curves observed comparing ZIKV-Dak infected IL-33- versus IL1rl1-deficient mice. Is this difference significant? These data suggest an alternative IL-33 receptor is involved in survival from ZIKV-Dak infection. Also, the trend towards decreased viral loads within the brainstem of IL1rl1-deficient mice infected with ZIKV-Dak should be further evaluated as this is consistent with the improved survival curve for this genotype compared with IL-33 KO mice. Importantly, a kinetic analysis of viral loads over time, in both rostral and caudal CNS, is required to conclude that loss of IL-33 or its receptors does not impact viral clearance. The difference in viral clearance within the CB by 10 dpi between WT and IL1rl1-deficient mice supports the need for this additional analysis.

Figure 2. It was very difficult to assess the data in this figure as no higher power images were provided and the differences in immunodetection of IL-33 is not that convincing – no conclusion can be drawn from 2 positive versus 0 positive cells. Also, Olig2 is expressed by oligodendrocyte precursor cells and by multipotential polydendrocytes, as is NG2, which could help identify these cells. In the HIP, about 76% of Olig2+ cells have NG2 co-expression, which is the highest among selected regions (doi.org/10.1186/s13041-021-00747-0). Plp immunostaining is a better marker of mature oligodendrocytes, which would also show their processes. The authors should perform double-label RNAscope studies using more appropriate cell markers to identify the cellular sources of IL-33 more convincingly. Please also denote the brain regions evaluated in each image. It is also confusing why the authors focus on areas of synapse elimination to evaluate myelinating cells, as synapses are not myelinated. Finally, the robust loss of survival phenotype observed in WNV-infected mice with IL-33 deletion in oligodendrocytes is striking. It is unclear why this result is not follow-up at all.

Figure 3. As mentioned by the author, clearance of i.c. infection with WNV-E218A relies on innate immune mechanisms, specifically the effects of interferon stimulated genes, such as IFIT1. Other papers cited show no role for T cells in viral clearance using this strain (Refs 19 and 15). Thus, this model is not appropriate to study the role of IL-33 in T cell-mediated virologic control. The authors should instead evaluate this in their ZIKV-Dak model, especially given the significant increase in T cell numbers and a trend of increased activation markers in CD8 T cells observed. It is unclear why the authors chose to pivot to T cells rather than following up on results examining glial cells.

Figure 4. At 7 dpi, monocytes heavily infiltrate the WNV-infected CNS and microglia become activated. As both express Iba1, this molecule cannot distinguish between these cells. Microglial specific markers should be used to distinguish effects on infiltrating versus resident myeloid cells. The authors cannot conclude effects specifically for microglia without these tools. Also, CX3CR1 is also expressed by T cells during viral infections

The correlations between IL-33 and Iba1 is poor. The data evaluating Iba1 at 10 dpi is interesting, but it is unclear whether there is loss of the macrophages or activated microglia. Microglial markers are needed to determine this. Similarly, the RNAseq experiment cannot distinguish between these two cell types. Finally, as CX3CR1 is expressed by memory T cells during viral clearance (10.1038/ncomms9306), the CreLox experiment does not specifically target microglia. The conclusion that ST2 expression on microglia is responsible for macrophage activation and host survival is not supported by the experimental approaches used.

Figure 5. In this figure the authors isolated myeloid cells for RNAseq. However, it is unclear whether the alterations in gene expression are due to effects on macrophages or microglia, or even on differential recruitment of myeloid cells. The authors need to address this issue throughout the manuscript.

Figure 6. The data evaluating myeloid cell survival are not very convincing as the flow cytometric analysis does not show a large effect and there are limited numbers of cells detected via immunostaining, which are unlikely to impact survival of the mice. Again, the lack of microglial markers does not allow the authors to conclude that activation and apoptosis are limited to microglia. Indeed, increased monocyte entry would mean more myeloid cell death as this commonly occurs in the setting of viral encephalitis. The IgG detection is not consistent with BBB disruption - the IgG would be localized strongly to vessels and fan out from them into the parenchyma. The increase in GFAP is consistent with immune activation of astrocytes, however, why was this not observed in FigS2?

Reviewer #2: Importantly, the authors conclude that microglia activation (Iba-1 MFI) and microglia apoptosis are influence by IL-33/ST2. In figure 6A, the number of dead microglia in the tissue prep are increased, but it is not clear (unless I missed it) whether live microglia number decreases in ST2 or IL-33 knockout mice. To this end, Iba-1 MFI is clearly decreased in knockout mice, but is this due to the absence of the cells or the lack of “activation.” The title of Figure 4 conveys the conclusion that microglia activation is reduced in the absence of IL-33 based primarily on Iba-1 intensity. Because this marker is also used to show the presence of cell (including some monocyte-derived cells which appear to increase in ko mice), it is difficult to support a conclusion for “activation” based on Iba-1 alone. Perhaps some “activation” genes from the RNAseq dataset could be helpful for further demonstrating this point. For example, reduced expression of MHC molecules, ccl2, or itgax and higher expression of p2ry12, sall1/3, or fcrls in microglia from knockout mice would greatly support this conclusion.

Reviewer #3: The resolution of Figures 2 to 6 is poor. As a result, I could not provide a detailed comment about the results presented in these figures. The quality of the images is not sufficient to observe positive staining as the authors claim. The same issue applies to figures without immunofluorescence (IF) images, as the legends are too small to read. Due to the

**Part III – Minor Issues: Editorial and Data Presentation Modifications**

Reviewer #1: (No Response)

Reviewer #2: 1) The authors nicely show a decrease in oligodendrocytes expressing IL-33, but there is also a decrease in IL-33+ astroyctes following infection (Figure 2C). This result should be mentioned in the results and discussion that astrocytes may also be a relevant source of IL-33.

2) ST2 is notoriously difficult to stain for using flow cytometry. The authors do not detect ST2 expression by flow in several cell types (Figures 3C and 3D), but nicely show Il1rl1 expression by real-time PCR. For example, Figure S4I nicely shows expression in microglia. Thus, examination of potential ST2-expressing population could be demonstrated by PCR, RNAscope, or other methods. Alternatively, the conclusions made from the flow cytometry staining could be discussed with the caveat that ST2 flow cytometry staining has rarely implicated cell types in the CNS that are important for responses to IL-33.

3) The authors include a description of similar studies performed with Toxo infection (line 84-86). The authors should correct the statement to reflect that astrocytes were the key responders to IL-33 in this study.

Reviewer #3: Lines 87 to 88 "whether IL-33 contributes to beneficial or pathogenic responses to neurotropic viral infection remains unexplored"

The authors must consider some literature reports such as DOI 10.1111/imm.12988 and especially DOI 10.1186/s12974-016-0628-1 in the context of neuroinflammation and IL-33 signaling.

Lines 95 to 98 "Specifically, during acute infection, microglia were shown to phagocytose neuronal processes the murine hippocampus, a phenomenon that limits viral dissemination while causing disease sequelae specific to episodic spatial memory"

This sentence seems odd, please review it carefully

Lines 103 to 106 "We found that following CNS infection oligodendrocyte numbers acutely decrease, with IL-33 expression by oligodendrocytes necessary for host survival, implicating release of IL-33 from dying oligodendrocytes."

Please review, I believe it refers to reduced IL-33 expression as a consequence of diminished numbers of oligodendrocyte cells

Lines 134 to 136 - Please explain the rationale for using IL-1 and IL-33 knockout mice to study type I IFN genes. ZIKV is also sensitive to type I IFN and I don't see the correlation here

PLOS authors have the option to publish the peer review history of their article (what does this mean?). If published, this will include your full peer review and any attached files.

Reviewer #1: No

Reviewer #2: No

Reviewer #3: **Yes: **Rafael Freitas Oliveira Franca
---

## [Decision Letter · Decision Letter 1]

5 Nov 2023

Dear Dr. Oberst,

We are pleased to inform you that your manuscript 'Oligodendrocyte-derived IL-33 functions as a microglial survival factor during neuroinvasive flavivirus infection' has been provisionally accepted for publication in PLOS Pathogens.

Best regards,

Kellie A. Jurado

Academic Editor

PLOS Pathogens

Alexander Gorbalenya

Section Editor

PLOS Pathogens

Kasturi Haldar

Editor-in-Chief

PLOS Pathogens

orcid.org/0000-0001-5065-158X

Michael Malim

Editor-in-Chief

PLOS Pathogens

orcid.org/0000-0002-7699-2064

Reviewer Comments (if any, and for reference):

Reviewer's Responses to Questions

**Part I - Summary**

Reviewer #2: The authors have addressed all the critiques from the prior review.

**Part II – Major Issues: Key Experiments Required for Acceptance**

Reviewer #2: The authors have addressed all the critiques from the prior review.

**Part III – Minor Issues: Editorial and Data Presentation Modifications**

Reviewer #2: The authors have addressed all the critiques from the prior review.

PLOS authors have the option to publish the peer review history of their article (what does this mean?). If published, this will include your full peer review and any attached files.

Reviewer #2: No

---

## [Editor Report · Acceptance letter]

15 Nov 2023

Dear Dr. Oberst,

We are delighted to inform you that your manuscript, "Oligodendrocyte-derived IL-33 functions as a microglial survival factor during neuroinvasive flavivirus infection," has been formally accepted for publication in PLOS Pathogens.

Best regards,

Kasturi Haldar

Editor-in-Chief

PLOS Pathogens

orcid.org/0000-0001-5065-158X

Michael Malim

Editor-in-Chief

PLOS Pathogens

orcid.org/0000-0002-7699-2064